# Multi-Objective Optimal Operation of Building Energy Management Systems with Thermal and Battery Energy Storage in the Presence of Load Uncertainty

**Parichada Trairat and David Banjerdpongchai \***
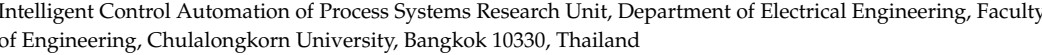

Intelligent Control Automation of Process Systems Research Unit, Department of Electrical Engineering, Faculty of Engineering, Chulalongkorn University, Bangkok 10330, Thailand
* Correspondence: bdavid@chula.ac.th

**Abstract:** This paper presents the optimal operation of a building energy management system (BEMS), with combined heat and power (CHP) generation, thermal energy storage (TES), and battery energy storage (BES), subject to load demand uncertainty. The main objective is to reduce the total operating cost (TOC) and total $CO_2$ emission (TCOE). First, we develop two models of load demand forecasting, one for weekday and the other for weekend, using artificial neural networks, long short-term memory, and convolutional neural networks. Then, we incorporate the predicted load demand and load demand uncertainty for planning the energy dispatch of the BEMS. TES aims to store the thermal energy waste from the power generation of *CHP* and discharge the thermal energy to the absorption chiller to supply the cooling load. BES and spinning reserve (SR) play an important role in handling the uncertainty of the load demand. The operation of BEMS, subject to the load demand uncertainty, is formulated as a linear program. We can efficiently solve the linear program and provide an optimal solution that satisfies the dispatch constraints. Thereafter, we determine the optimal size of BES, based on economics and environmental optimal operation. The proposed BEMS is compared to the previous BEMS, without BES and SR. Furthermore, we propose the multi-objective optimal operation, where the normalization for *TOC* and *TCOE* is introduced, and the multi-objective function is defined as a linear combination of normalized *TOC* and *TCOE*. The numerical results reveal the trade-off relationship between *TOC* and *TCOE*. In particular, when *TCOE* is minimum, *TOC* becomes maximum. On the other hand, when *TOC* is minimum, *TCOE* becomes maximum. The relationship provides a method to select the operating point, as well as analyze the power flow for the multi-objective optimal operation.

**Keywords:** combined heat; power generation; thermal energy storage; battery energy storage; optimal dispatch; building energy management system; load forecasting; load demand uncertainty; multi-objective approach

## 1. Introduction

The energy management system is one of the control systems that aims to dispatch energy with efficiency, according to demand. Nowadays, in large buildings, such as shopping malls, offices, commercial buildings, or hotels, the energy management system has been applied. There are many types of energy needs in buildings, such as electric, cooling, or heat energy. For supply, the generation system can produce the energy, or the operator can purchase energy from the power grid. In specific important operations, energy sources in the building consist of combined heat and power (CHP) as the main power source [1]. In [2], the BEMS has *CHP* as the main power source and works in conjunction with TES, which stores the waste heat generated by *CHP* in the electric power production process and aims to reduce total operating cost and $CO_2$ emission. Likewise, Ref. [3] proposed the optimal operation for reducing carbon emissions by using a two-layer model for AC-DC hybrid microgrids. Carbon trading mechanisms and uncertainty are considered.

To plan for energy dispatch, load demand prediction plays an important role in demand management and making distribution planning as efficient as possible. For predicting load demand, there are a variety of methods, such as neural networks, unit consumption methods, and trend forecasting methods, which can be applied according to the amount of data and desired period of time for prediction. There are research reports on load demand prediction, which takes the load demand forecast error into consideration when planning the power distribution. The prediction error is classified in the form of uncertainty. Various prediction methods are widely used. In [4], the load is predicted, and the uncertainty is estimated using long short-term memory for a day-ahead load of EV charging stations. There are research works taking predictions into consideration when planning energy distribution. In [5], predicting the load demand with an artificial neural network model is used to plan energy distribution and reduce the operating costs of a smart house. Modelling load demand forecast errors can adjust dispatch strategies whenever the hourly power load demand changes. Likewise, Ref. [6] proposed the energy dispatch strategy to schedule the charging and discharging of the battery-integrated power system. The fluctuating behavior of electrical loads was analyzed to find a suitable strategy by taking the battery characteristics and system parameters into consideration. A robust energy management system is an efficient energy management system to handle load uncertainty in the system. For a small system, such a microgrid with renewable energy, the uncertainty of the power source is important to consider. In [7], the authors proposed the robust energy management system to deal with the uncertainty using the dual decomposition and find the optimal solution that reduces the operating cost. Similarly, in the stand-alone microgrid, the errors from predicting the power generation, including solar power, wind power, and the electricity load demand of local electricity users, are taken into account for planning the schedule of the microgrid supply [8]. For cluster microgrids, the prediction of the load demand in the short-term using ANN was used to manage the operations [9].

There are various sources of uncertainty occurring within the system, such as renewable energy and electrical load demand. This uncertainty is difficult to control, but it is important to deal with the uncertainty that arises in load demand predictions. In this paper, we propose the multi-objective optimal operation of a building energy management system (BEMS) with thermal energy storage (TES) and battery energy storage (BES), with consideration of the load demand uncertainty. Moreover, we will find the trade-off performance between the operating cost and carbon dioxide emission and analyze the power flow. The proposed BEMS is applicable to large commercial buildings with electrical, cooling, or thermal load demands. Large commercial buildings include shopping malls, office buildings, hotels, or factories. We consider the electrical and thermal energy supply and consumption for efficiency and comfort. The ratio between electrical and thermal energy is concerned with electrical energy from *CHP* and BES and thermal energy from *CHP*. The ratio plays a vital role in the power-to-heat (*P2H*) coefficient. In this work, load characteristics consist of time variation and uncertainty. The proposed BEMS can accommodate large buildings subject to the load variation. For planning the dispatch strategies, the load uncertainty is taken into account. The proposed BEMS has *CHP* as the main generating component. *CHP* typically achieves a total efficiency of 65 to 80%. *CHP* requires less fuel to produce energy output and depends on a natural gas source. Typically, *CHP* range in size from 50 kW to over 1 MW electrical capacity [10]. Therefore, the size of *CHP* can be chosen to suit the power and heat consumption of the building. The proposed BEMS incorporates TES and BES, which manage the excess heat energy and electricity to be used during the on-peak period. Moreover, the uncertainty of the load demand is taken into account in the energy dispatch of the proposed BEMS. The optimal strategy is designed to accommodate the load uncertainty and provide economic and environmental benefits to users. We demonstrate that the improvement of *TOC* using the proposed BEMS occurring in the case of economic optimal operation and multi-objective operation in the presence of load uncertainty. This approach is universal for the types of loads with uncertainty.

The paper is organized as follows. In Section 2, we describe the BEMS. Section 3 presents load demand forecasting and load uncertainty. Section 4 provides the formulation of the optimal dispatch of the proposed BEMS. Section 5 shows the comparison results, in terms of *TOC* and *TCOE* and energy flow. The conclusion is provided in Section 6.

## 2. BEMS Description

The proposed system consists of *CHP*, absorption chiller, auxiliary boiler, TES, BES, and power grid. *CHP* is the main component of power generation for distribution by producing electrical energy and heat energy simultaneously, together with the electric power from the BES and SR. BES and SR are available to operate when load demand uncertainty arises, and there is a power grid that will help protect them in the event of a power shortage. The proposed BEMS diagram is shown in Figure 1. In addition, *CHP* makes profits from exchanging electrical energy to the grid. Moreover, an absorption chiller is an element that converts heat energy into cooling energy to supply cooling load demands within a building. The chiller has a heat source from the *CHP*, TES, and auxiliary boiler. From the electrical power generation process of *CHP*, the waste heat is stored in TES for further use. The description of the variables is shown in Table A1 in the Appendix A.

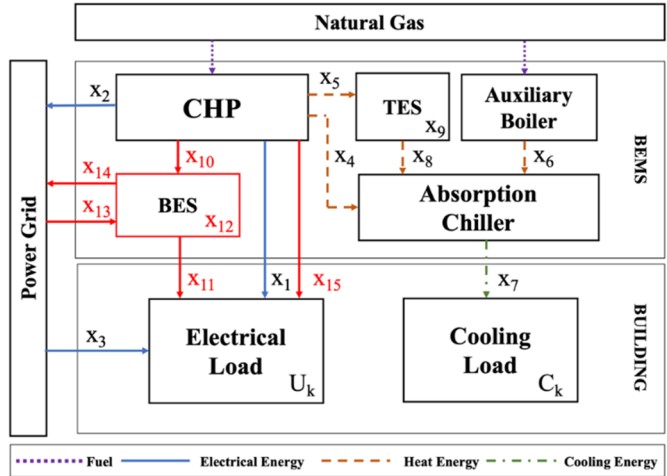

**Figure 1.** Diagram of the proposed BEMS.

## 3. Load Demand Forecasting and Uncertainty

From the considered BEMS, inside the building consists of two types of energy demands, electrical energy and cooling energy, to plan for efficient energy dispatch and see the trends in energy consumption within the building [11]. Predictions of electrical load demand have been used to analyze the uncertainty of the electrical load demand and create a set of load demands under load demand uncertainty to be used in energy dispatch strategies. A total of three prediction methods were used. They consist of artificial neural network (ANN), long short-term memory (LSTM), and convolutional neural network (CNN).

The load demand prediction is intended to determine the amount of load in the future times, while minimizing the predicted error value. The resulting error values are taken into account to create a set of uncertainties and used in energy dispatch planning. In this work, three learning models were used to predict load demand. We consider the electrical load profiles of a large shopping mall in Bangkok, Thailand, as loads for BEMS. The cooling load demand was modified from real electrical load profiles. The data have a sampling time of 15 min. and cover an operation of 1 month. The shopping mall utilizes electricity from the Metropolitan Electricity Authority (MEA), with a 69-kV distribution grid [10].

### 3.1. Forecasting Models

The prediction is predicted 1 step in advance, and the predicted electrical load demand is represented by the following equation

$$\widehat{U}_k = U_k + \Delta U_k \tag{1}$$

where $\widehat{U}_k$ represents the predicted load. $U_k$ is the actual load, and $\Delta U_k$ is the difference between the actual and predicted loads, $U_k - \widehat{U}_k$. The data used to predict load demand is the data set for the electricity load demand of large shopping malls in megawatts per hour for a period of time. The data is divided into two sets: the weekday and weekend sets. The training and testing sets accounted for 75% and 25% of the data sets, respectively. The input data includes load demand, 1–4 h historical load demand, load demand on the previous day same hour, load demand on the previous week same hour, hour of the day, and day of the week. The output is the predicted load in the next step.

#### 3.1.1. Artificial Neural Network Model

The neural network model is a developed model that mimics the basic biological nervous system. The key component of a neural network is a large number of interconnected neurons. The data is fed into the neural network for training with two hidden layers. As shown in Figure 2, the number of neurons in each hidden layer for weekdays is represented by *n*. Figure 3 shows the structure for weekends with the number of neurons in each hidden layer, represented by m. Both models are taught using a Bayesian regularization backpropagation algorithm. The proposed artificial neural network has nine inputs. The best prediction outcome is obtained when the number of neurons for the weekday model is *n* = 10 and number of neurons for the weekend model is m = 3. After predicting the load demand, Figure 4 shows the load profile between the actual and predicted load demand.

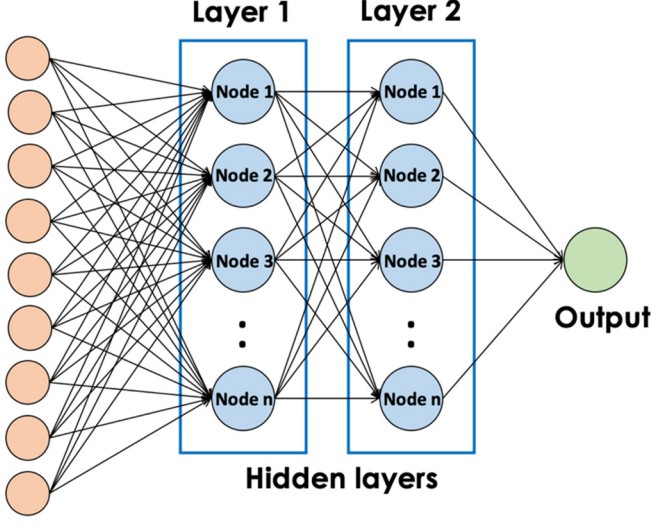

**Figure 2.** Structure model of artificial neural network for weekday load.

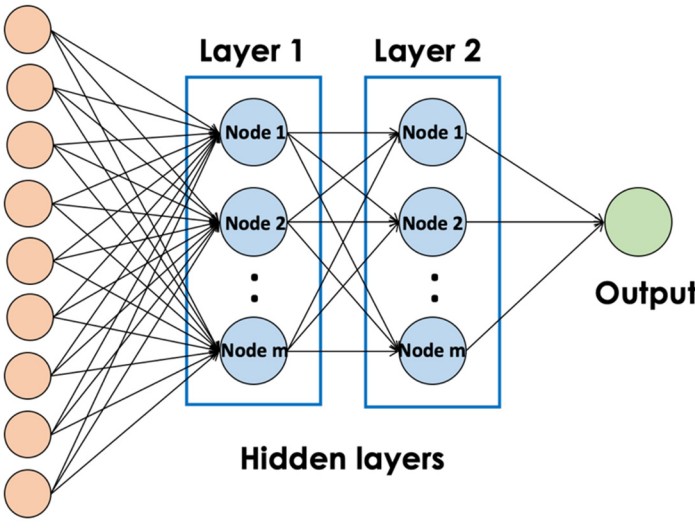

**Figure 3.** Structure model of artificial neural network for weekend load.

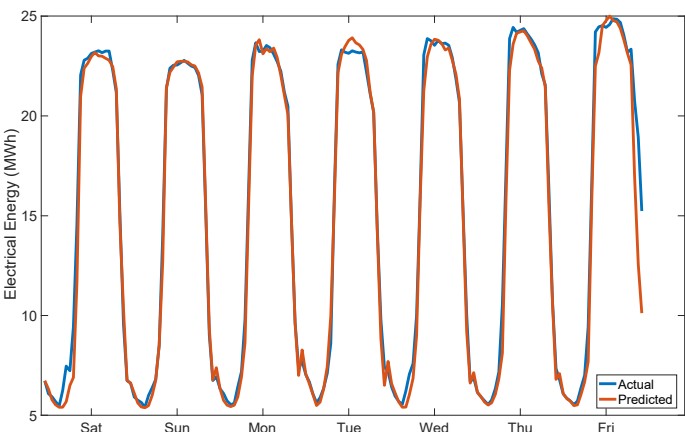

**Figure 4.** Actual load demand and predicted load demand from ANN model.

### 3.1.2. Long Short-Term Memory Model

Long short-term memory is a type of neural network designed for sequential processing. This is a type of recurrent neural network because the outgoing loop is used in processing. The long short-term memory model has an additional part of memory that is effective in decision-making for reading, writing, and erasing [12]. The proposed long short-term memory model consists of nine inputs, defined as a sequence input layer, three long short-term memory layers, a fully connected layer, and a regression output layer using the adaptive moment estimation training option. Structural model of long short-term memory model is shown in Figure 5, and the load profile of actual load and predicted load, shown in Figure 6.

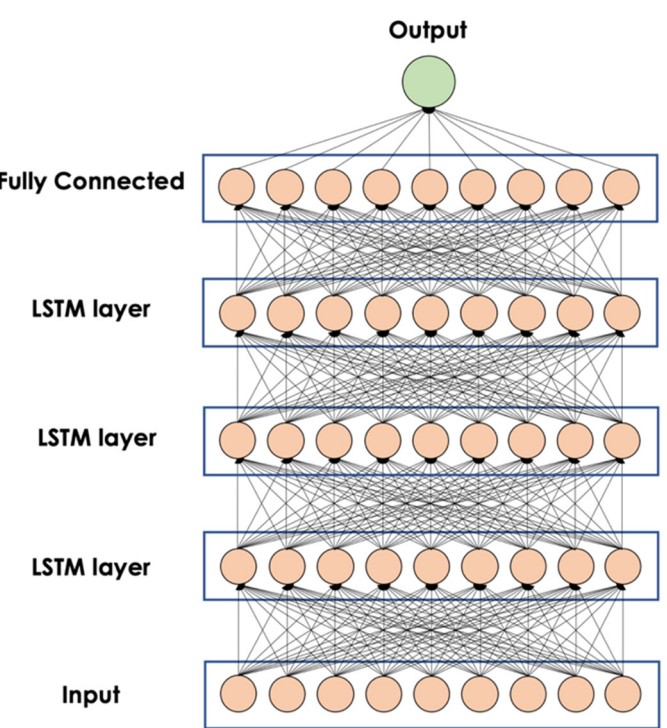

**Figure 5.** Structure model of long short-term memory.

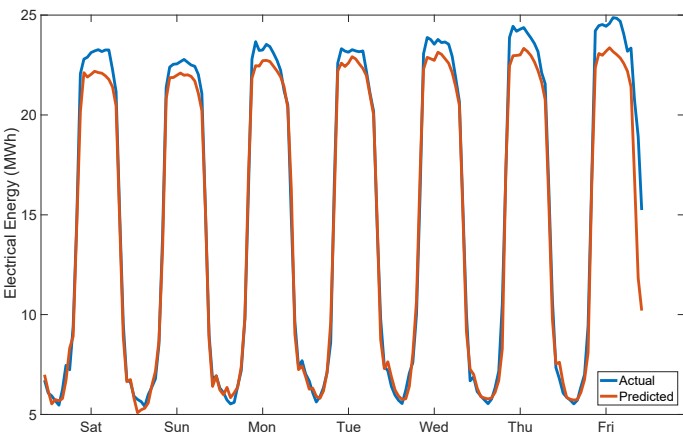

**Figure 6.** Actual load demand and predicted load demand from LSTM model.

### 3.1.3. Convolutional Neural Network Model

The convolutional neural network model is a type of neural network model. It simulates human vision, divides it into sub-areas, and brings the group of sub-areas to merge to process. To process, the key component of the convolutional neural network model is the filter. Generally, one filter can extract one type of attribute, and the models can range from 1- to multi-dimensional. The structure of CNN model consists of nine inputs, three one-dimensional convolution layer, one fully connected layer, and the output with the adaptive moment estimation training option. The size and number of filters are 5 and 150, respectively. The model of convolutional neural network model is shown in Figure 7, and the load profile of the actual load and predicted load is shown in Figure 8.

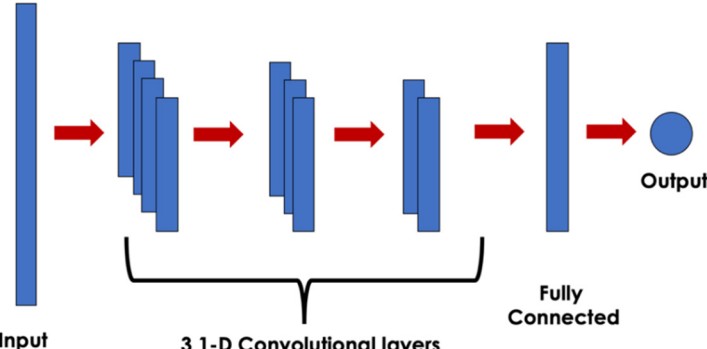

**Figure 7.** Structure model of convolutional neural network.

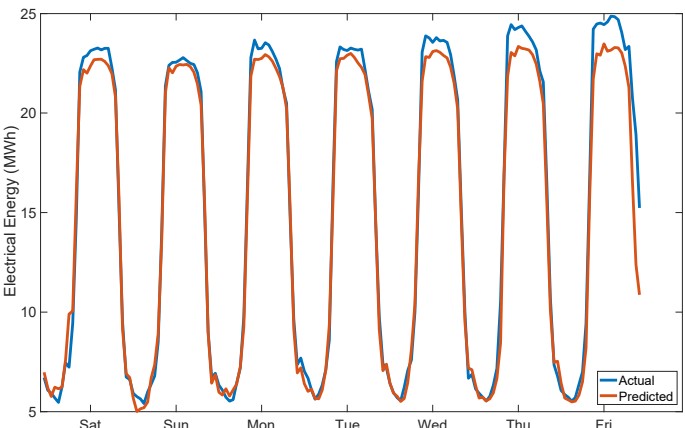

**Figure 8.** Actual load demand and predicted load demand from CNN model.

### 3.2. Performance Measurement

To measure the performance of each predictive model, error calculations were taken to evaluate performance using the following measurements.

#### 3.2.1. Mean Absolute Percentage Error

Mean absolute percent error (MAPE) is a measure of the predictive accuracy of a statistical prediction method. The precision is expressed as a ratio, given by the following equation.

$$\text{MAPE} = \frac{100\%}{n} \sum_{k=1}^{n} \left| \frac{U_k - \widehat{U}_k}{U_k} \right| \tag{2}$$

#### 3.2.2. Root Mean Square Error

Root mean square error (RMSE) is a measure of the difference between the frequently used sample or population values predicted by a model or estimator and the observed values, as determined by the following equation.

$$\text{RMSE} = \sqrt{\frac{\sum_{k=1}^{n} \left( \widehat{U}_k - U_k \right)^2}{n}} \tag{3}$$

The predicted load demands of each model were compared by MAPE and RMSE to find the model with the best predictive performance and most accuracy (of the three models). The data based on electrical load of the large shopping mall for 7 days and separate to weekday and weekend.

We compare the forecasting errors of the load demand from three models, namely ANN, LSTM, and CNN in Table 1. For ANN, the results show that a predicted load closely

follows the actual load. During high load demands, there is little error in prediction. It is observed that LSTM and CNN have relatively large errors in the high demand range. All models have discrepancies in the last few hours of the prediction. We utilized MAPE and RMSE to evaluate the prediction performance. It can be seen Table 1 that the MAPE and RMSE of ANN are lower than that of LSTM and CNN. Thus, ANN gives the most accurate predictions. Subsequently, we took the prediction errors into account in the energy dispatch. In order to accommodate the load demand uncertainty, we employed BES and SR to support the load demand uncertainty. The load demand prediction and uncertainty reveal the trends of energy consumption in the building and serve as the important input data for the energy dispatch.

**Table 1.** Forecasting errors of forecasting models.

| Measure | ANN | LSTM | CNN |
|---|---|---|---|
| | Weekday | | |
| MAPE | 3.93 | 4.21 | 5.95 |
| RMSE | 1.01 | 1.09 | 1.38 |
| | Weekend | | |
| MAPE | 3.75 | 3.92 | 4.44 |
| RMSE | 0.72 | 0.59 | 0.72 |

### 3.3. Worst-Case Error and Uncertainty

After predicting the electrical load demand with an artificial neural network model, we analyzed the difference between the actual and predicted load demand to create the uncertainty of the predicting electrical load demand. Worst-case error is defined as the maximum error of load demand forecasting for weekdays or weekends for each period ($k = 0, 1, 2, \ldots, 23$) and represented by $\Delta U_{k,max}$. The worst-case error of load demand forecasting is shown in Figure 9.

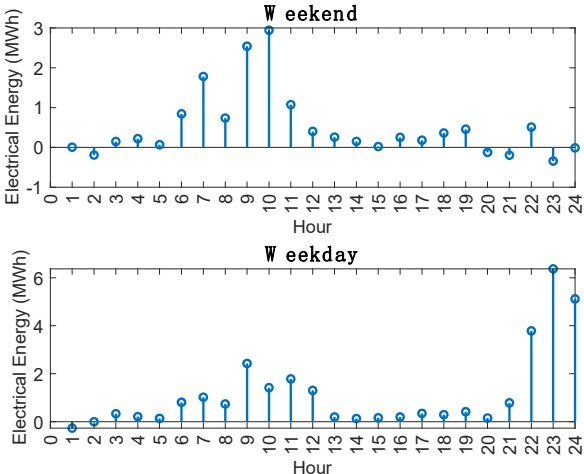

**Figure 9.** The worst-case error of load demand forecasting.

In order to plan energy distribution by backing up production capacity, in accordance with the demand for electricity, and maintain a balance between electricity generation and consumption, the uncertainty set [13] consists of the worst error between the actual and predicted load demand for each period. The worst errors were taken into account on the positive side. For cases where the worst error was negative, we set the uncertainty to have a value of zero. The uncertainty of load demand forecasting is shown in Figure 10.

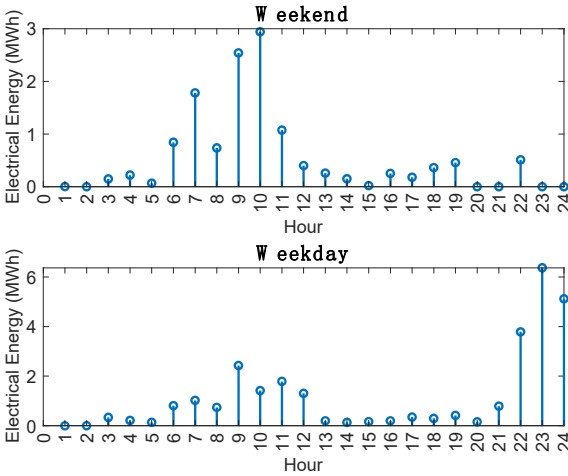

**Figure 10.** The uncertainty of load demand forecasting.

After obtaining the uncertainty of load demand forecasting, the maximum uncertainty was taken into account in the predicted load demand. We defined the predicted load under the worst-case uncertainty, represented by $\widehat{U}_{wc,k}$, as follows.

$$\widehat{U}_{wc,k} = U_k + \Delta U_{k,max} \tag{4}$$

It is noted that $\widehat{U}_{wc,k}$ will be used to plan a strategy for the energy dispatch in the Section 4.

## 4. Optimal Dispatch Strategies

To determine the appropriate dispatch strategy, the specific objectives were considered, in order to make the strategy effective and meet the demands. We formulate the problem of finding an optimal strategy for energy management in a building.

### 4.1. Dispatch Strategies

For the dispatch strategy of the proposed BEMS, two objectives will be considered. They are economics optimal operation and environmental optimal operation. After that, both objectives are simultaneously considered, namely multi-objective optimal operation. There are two types of energy needed in buildings, namely electrical and cooling loads. Energy dispatch strategies are divided into two categories, based on the type of load demand. In the BEMS, thermal energy storage (TES) and battery energy storage (BES) are installed. Moreover, an additional amount of energy from *CHP* is called the spinning reserve (SR) to tackle the load uncertainty. Next, we present the characteristics of the elements and dispatch strategies. Note that the description of system parameters and acronyms are given in Appendix A, Tables A2 and A3, respectively.

#### 4.1.1. Thermal Energy Storage

TES is a device for temporarily retaining thermal energy for later use. The waste heat from the electrical production process of *CHP* will be stored in TES. The constraints of the TES consist of the rate of charge and discharge, state-of-charge, and operating boundary between the minimum and maximum capacity. The dispatch of TES is as follows.

$$\varepsilon x_{5,k} \leq R_1 \tag{5}$$

$$\frac{1}{\delta}x_{8,k} \leq R_2 \tag{6}$$

$$x_{9,k} = init(1-\mu)^k + \sum_{j=1}^{k}\left[(\varepsilon x_{5,j}) - (\frac{1}{\delta}x_{8,j})\right](1-\mu)^{k-j+1} \tag{7}$$

$$S_{min} \leq x_{9,k} \leq S_{max} \tag{8}$$

### 4.1.2. Battery Energy Storage

BES is the storage of electrical energy to be used during the on peak period and support the BEMS during the load uncertainty. BES will charge the energy during the first four hours of the day and discharge energy to support the uncertainty which arises. The constraints for BES include the charge and discharge rates, state-of-charge, and operating boundary between minimum and maximum capacity. The dispatch of BES is as follows.

$$0 \leq x_{10,k} + x_{13,k} \leq d_{sc} \tag{9}$$

$$0 \leq x_{11,k} + x_{14,k} \leq d_{sd} \tag{10}$$

$$x_{12,k} = x_{12,k-1} + \eta_c \eta_i (x_{10,k} + x_{13,k}) - \frac{(x_{11,k} + x_{14,k})}{\eta_c \eta_i} \tag{11}$$

$$B_{min} \leq x_{12,k} \leq B_{max} \tag{12}$$

### 4.1.3. Spinning Reserve

SR is the energy from *CHP* to support the uncertainty of the electrical load, together with BES. SR can be an additional energy resource when the generation does not exceed the maximum capacity of *CHP* [14]. SR works in conjunction with the BES to support the load demand uncertainty. The dispatch of SR is as follows.

$$P_{CHP,min}\Delta t \leq x_{1,k} + x_{2,k} + x_{10,k} + x_{15,k} \leq P_{CHP,max}\Delta t \tag{13}$$

$$\frac{x_{1,k} + x_{2,k} + x_{10,k} + x_{15,k}}{x_{4,k} + x_{5,k}} = P2H \tag{14}$$

$$\begin{aligned} |(x_{1,k} + x_{2,k} + x_{10,k} + x_{15,k}) \\ (x_{1,k-1} + x_{2,k-1} + x_{10,k-1} + x_{14,k-1} + x_{15,k-1})| \leq R_{CHP} \end{aligned} \tag{15}$$

### 4.1.4. Electrical Energy

For the strategy to dispatch the electricity, a set of predicted load demands are planned in the energy dispatch. It was divided into four cases, based on predicted load, under load demand uncertainty. *CHP* is the main components to supply electrical power to the electric load. The first case is when there is no demand for electrical loads, and all components in the system will shut down. Regarding the second case, when the actual load demand is less than the predicted load demand, *CHP* cooperates with BES to supply the energy to meet demands. In the third case, the actual load demand is less than the predicted load demand and maximum capacity of *CHP*. *CHP* supplies electrical energy equal to the predicted load demand. BES cooperate with SR to support part of the uncertainty. The last case is when the actual load demand is less than the predicted load demand and more than the maximum capacity of *CHP*. *CHP* supplies electrical energy equal to the predicted load demand. BES cooperate with SR and purchase electrical energy from power grid to support the uncertainty. The electrical energy dispatch strategy is shown below.

If $U_k = 0$, then

$$x_{1,k} = x_{10,k} = x_{14,k} = x_{4,k} = x_{5,k} = 0$$

Else if

$$U_k \leq \widehat{U}_{wc,k} \ \&\& \ U_k \leq P_{CHP,max}$$

Then

$$P_{CHP,min}\Delta t \leq x_{1,k} + x_{2,k} + x_{10,k} \leq P_{CHP,max}\Delta t$$

$$\frac{x_{1,k} + x_{2,k} + x_{10,k}}{x_{4,k} + x_{5,k}} = P2H$$

$$|(x_{1,k} + x_{2,k} + x_{10,k}) - (x_{1,k-1} + x_{2,k-1} + x_{10,k-1})| \leq R_{CHP}$$

$$x_{12,k} = x_{12,k-1} + \eta_c \eta_i (x_{10,k} + x_{13,k}) - \frac{(x_{11,k} + x_{14,k})}{\eta_c \eta_i}$$

$$S_{min} \leq x_{12,k} \leq S_{max}$$

If $k \leq 4$, then

$$x_{1,k} \leq U_k$$

$$x_{10,k} \leq d_{sc}$$

$$0 \leq x_{10,k} \leq d_{sc}$$

Else

$$x_{1,k} + x_{11,k} \leq U_k$$

end.
Else if

$$U_k > \widehat{U}_{wc,k} \,\&\&\, U_k < P_{CHP,max}$$

Then

$$P_{CHP,min}\Delta t \leq x_{1,k} + x_{2,k} + x_{10,k} + x_{15,k} \leq P_{CHP,max}\Delta t$$

$$\frac{x_{1,k} + x_{2,k} + x_{10,k} + x_{15,k}}{x_{4,k} + x_{5,k}} = P2H$$

$$|(x_{1,k} + x_{2,k} + x_{10,k} + x_{15,k}) - (x_{1,k-1} + x_{2,k-1} + x_{10,k-1} + x_{14,k-1} + x_{15,k-1})| \leq R_{CHP}$$

$$x_{12,k} = x_{12,k-1} + \eta_c \eta_i (x_{10,k} + x_{13,k}) - \frac{(x_{11,k} + x_{14,k})}{\eta_c \eta_i}$$

$$S_{min} \leq x_{12,k} \leq S_{max}$$

If $k \leq 4$, then

$$x_{1,k} \leq \widehat{U}_{wc,k}$$

$$x_{15,k} = U_k - \widehat{U}_{wc,k}$$

$$x_{10,k} \leq d_{sc}$$

$$0 \leq x_{10,k} \leq d_{sc}$$

Else

$$x_{1,k} \leq \widehat{U}_{wc,k}$$

$$x_{11,k} + x_{15,k} = U_k - \widehat{U}_{wc,k}$$

end.
Else

$$P_{CHP,min}\Delta t \leq x_{1,k} + x_{2,k} + x_{10,k} + x_{15,k} \leq P_{CHP,max}\Delta t$$

$$\frac{x_{1,k} + x_{2,k} + x_{10,k} + x_{15,k}}{x_{4,k} + x_{5,k}} = P2H$$

$$|(x_{1,k} + x_{2,k} + x_{10,k} + x_{15,k}) - (x_{1,k-1} + x_{2,k-1} + x_{10,k-1} + x_{14,k-1} + x_{15,k-1})| \leq R_{CHP}$$

$$x_{12,k} = x_{12,k-1} + \eta_c \eta_i (x_{10,k} + x_{13,k}) - \frac{(x_{11,k} + x_{14,k})}{\eta_c \eta_i}$$

$$S_{min} \leq x_{12,k} \leq S_{max}$$

If $k \leq 4$, then

$$x_{1,k} \leq \widehat{U}_{wc,k}$$

$$x_{15,k} = U_k - \widehat{U}_{wc,k}$$

Else

$$x_{1,k} \leq \widehat{U}_{wc,k}$$

$$x_{3,k} + x_{11,k} + x_{15,k} = U_k - \widehat{U}_{wc,k}$$

end.

Note that *CHP* does not operate beyond the boundary of the power output and ramp rate. Similarly, BES does not operate beyond the boundary of capacity, and the state-of-charge is updated every time instant $k$.

4.1.5. Cooling Energy

The cooling energy dispatch strategy is divided into four cases, as well as the electric energy dispatch strategy. The main components of cooling energy dispatch include *CHP*, auxiliary boiler absorption chiller, and TES. In the first case, there is no load demand of cooling energy. The second case is that the cooling load demand is less than the minimum operating value of the absorption chiller. *CHP* and TES work together to supply heat to the absorption chiller, and the absorption chiller operates at the minimum value of the machine rating. In the third case, the cooling load demand is greater than the minimum operating value of the absorption chiller, and *CHP* can provide sufficient heat energy. *CHP* cooperates with TES to supply heat to the absorption chiller. In the last case, *CHP* cannot provide enough heat energy. TES cooperate with auxiliary boiler to supply heat energy to the absorption chiller. The cooling energy dispatch strategy is shown below.

If $C_k = 0$, then

$$x_{4,k} = x_{6,k} = x_{7,k} = x_{8,k} = 0$$

$$x_{9,k} = (x_{9,k-1} + x_{5,k})(1 - \mu)$$

$$\varepsilon x_{5,k} \leq R_1 \Delta t$$

Else if $C_k \leq CP_{AC,min}\Delta t$, then

$$(x_{4,k} + x_{8,k})COP_{AC} = x_{7,k}$$

$$\frac{1}{\delta}x_{8,k} \leq R_2\Delta t$$

$$x_{9,k} = (x_{9,k-1} + x_{8,k})(1 - \mu)$$

$$x_{5,k} = x_{6,k} = 0$$

$$x_{7,k} = CP_{AC,min}\Delta t$$

Else if $C_k \leq \frac{P_{CHP,max}\Delta t}{P2H}COP_{AC}\Delta t$, then

$$(x_{4,k} + x_{8,k})COP_{AC} = x_{7,k}$$

$$\frac{1}{\delta}x_{8,k} \leq R_2\Delta t$$

$$x_{9,k} = (x_{9,k-1} + x_{8,k})(1 - \mu)$$

$$x_{5,k} = x_{6,k} = 0$$

$$x_{7,k} = min(C_k \frac{P_{CHP,max}\Delta t}{P2H}COP_{AC})$$

Else

$$(x_{4,k} + x_{6,k} + x_{8,k})COP_{AC} = x_{7,k}$$

$$\frac{1}{\delta}x_{8,k} \leq R_2\Delta t$$

$$x_{9,k} = (x_{9,k-1} + x_{8,k})(1 - \mu)$$

$$H_{AB,min}\Delta t \leq x_{6,k} \leq H_{AB,max}\Delta t$$

$$x_{5,k} = 0$$

$$x_{7,k} = min(C_k, C_{AC,max}(\frac{P_{CHP,max}\Delta t}{P2H} + H_{AB,max}\Delta t)COP_{AC})$$

end.

Note that $CP_{AC,min}$ and $CP_{AC,max}$ are the minimum and maximum of the cooling production of the absorption chiller. $COP_{AC}$ is the coefficient of the performance of a single-type absorption chiller. $H_{AB,min}$ and $H_{AB,max}$ are the minimum and maximum of heat energy production of auxiliary boiler.

### 4.2. Economics and Environmental Optimal Operations

In the proposed BEMS, the objective functions are divided into economics optimal operation and environmental optimal operation. Both objectives are subjected to energy dispatch conditions, i.e., electrical and cooling loads.

### 4.2.1. Economics Optimal Operation

Economics optimal operation defines the objective function to be the *TOC*. The *TOC* is the summation of the energy and demand charge costs [15]. The demand charge cost depends on the maximum imported electricity from the power grid. The objective is to minimize *TOC*.

$$TOC = EC + DCC \tag{16}$$

$$EC = \sum_{k=1}^{n \times d} C_{CHP}(x_{1,k} + x_{2,k} + x_{10,k} + x_{15,k}) + p_k(x_{3,k} + x_{13,k}) \\ -q_k(x_{2,k} + x_{14,k}) + C_{AB}x_{6,k} \tag{17}$$

$$DCC = \frac{d_{PG}}{\Delta t} \max_{h=1,...,n \times d} x_{3,k} \tag{18}$$

Note that $x_{i,k}$ is the energy flow. $q_k$ and $p_k$ represent the price of electrical energy exporting and importing to grids. $C_{CHP}$ and $C_{AB}$ are operating costs of the *CHP* and auxiliary boiler, respectively, and depend on fuel price. Moreover, $d_{PG}$ is the demand charge from power grids. $\Delta t$ is the time duration of each time interval, $n$ is the number of time interval in one day, and d is the number of days.

### 4.2.2. Environmental Optimal Operation

Environmental optimal operation defines the objective function to be the total $CO_2$ emissions. It is calculated from the $CO_2$ emissions in the energy production of the components in the system, with the aim of minimizing the total $CO_2$ emissions.

$$TCOE = EF_{CHP}(x_{1,k} + x_{2,k} + x_{10,k} + x_{15,k}) + GEFx_{3,k} + \frac{EF_{AB}}{\eta_{AB}}x_{6,k} \tag{19}$$

Note that $EF_{CHP}$ represents the $CO_2$ emission of *CHP* in ton per megawatt-hour, $EF_{AB}$ represents the $CO_2$ emission of auxiliary boiler in ton per megawatt-hour, $GEF$ is the $CO_2$ emission of power grid in ton per megawatt-hour, and $\eta_{AB}$ is the efficiency of the auxiliary boiler.

### 4.3. Multi-Objective Optimal Operation

From Section 4.2, there are two different objective functions, which are economics optimal operation and environmental optimal operation. To find the relationship between the two objectives, the multi-objective optimal operation is introduced. The relationship between the two objectives is represented by the weighting factor $\alpha$ and provided in the form of trade-off performance. The dispatch strategy of multi-objective optimal operation using the same energy dispatch strategy as described in Section 4.1.

In order to keep the cost function at the same unit of measure, we apply the normalization to the objective functions. The method is referred to the min–max normalization. This method is commonly used to normalize the cost function, where the lowest value is converted to 0 and highest value is converted to 1; all other values are converted to values between 0 and 1. Both objective functions will be normalized. The normalized total operating cost is represented by $J_{TOC}$, and the normalized total $CO_2$ emission is $J_{TCOE}$. The normalization of *TOC* and *TCOE* are expressed as follows:

$$J_{TOC} = \frac{TOC - TOC_{min}}{TOC_{max} - TOC_{min}} \tag{20}$$

$$J_{TCOE} = \frac{TCOE - TCOE_{min}}{TCOE_{max} - TCOE_{min}} \tag{21}$$

where $TOC_{min}$ is the minimum *TOC*, $TOC_{max}$ is the maximum *TOC*, $TCOE_{min}$ is the minimum total $CO_2$ emission, and $TCOE_{max}$ is the maximum *TCOE*. From the normalized *TOC* and *TCOE*, we can combine them into a multi-objective function *J*, defined as follows.

$$J = (1 - \alpha)J_{TOC} + \alpha J_{TCOE} \tag{22}$$

where $\alpha$ is the weighting factor between 0 and 1 to represent the weight between *TOC* and *TCOE* and have a step size of 0.1. The weights are 0 is the case, which considers only the economics optimal operation, and the weight is equal to 1, when considering only the environmental optimal operation.

For the uncertainty case, the minimum *TOC* is the *TOC* from economics optimal operation, and the minimum *TCOE* is the result of environmental optimal operation. On the other hand, the maximum of *TOC* is the operating cost of the environmental optimal operation, and the maximum value of *TCOE* is the $CO_2$ emission of economics optimal operation. For the nominal case, the energy dispatch is based on the predicted load, without considering the load uncertainty. The minimum *TOC* comes from the economics optimal operation of the nominal load. The minimum *TCOE* is obtained from the environmental optimal operation of the nominal load. The maximum *TOC* is from the environmental optimal operation result. The maximum *TCOE* is the result of the economic optimal operation.

### 4.4. Linear Programming and Algorithm

Linear programming is an optimization problem with a linear cost function subjected to the linear constraints. The operations of the proposed BEMS in the presence of the uncertainty in the electrical load demand can be formulated as linear programs. The linear program is described as follows.

$$
\begin{aligned}
minimize \quad & c^T x \\
subject\ to \quad & A_{eq}x = B_{eq} \\
& C_{ineq}x = D_{ineq} \\
& x \geq 0
\end{aligned}
$$

where *c* is the constant vector related to the cost function. The cost function is either *TOC* or *TCOE*. *x* is the decision variable, expressed as the energy flow, as shown in BEMS diagram. The constraints consist of the dispatch conditions of each component in BEMS, as described in Section 4.1. The optimization problem is defined as the minimization problem under the constraints of the dispatching condition of BEMS. The constraints are in the form of equations and inequalities to define operating conditions and boundaries. The optimal solution is the most suitable answer out of the feasible answers.

## 5. Numerical Results

### 5.1. Design of BES Sizing

The design of the BES sizing is to choose the proper size of the BES for the desired application of BEMS. It is the required size of BES, taking into account the optimal economic operation and environmental operation. Therefore, users can choose the size of BES for their intended purpose.

To determine the proper size of BES, we consider the most suitable economic operation of BES, with batteries of 2 to 6 megawatt-hour. The energy dispatch strategies in Section 4 are used to determine the minimum *TOC*. Figure 11 shows that the minimum *TOC* is obtained when the size of BES is 4.2 megawatt-hour. The corresponding *TOC* is 6,018,029 baht.

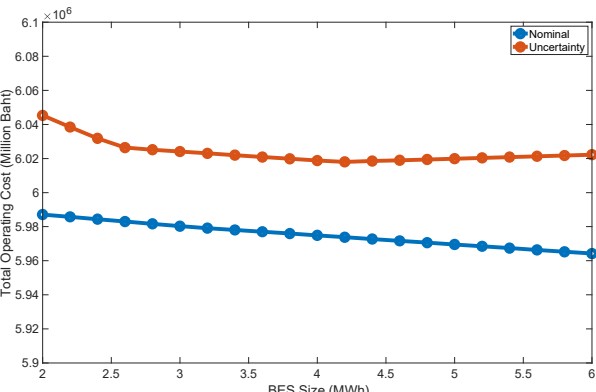

**Figure 11.** Comparison between BES size and *TOC*.

On the other hand, we experiment by varying the sizes of BES to determine the proper size, with the minimum $CO_2$ emission, using the energy dispatch strategy in Section 4.1. In Figure 12, the size of the BES that has the minimum *TCOE* is 2.5 megawatt-hour, with *TCOE* equal to 1565.11 tonCO$_2$.

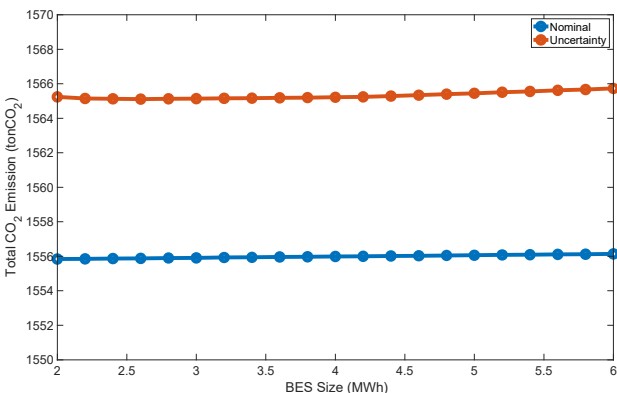

**Figure 12.** Comparison between BES size and *TCOE*.

From the design of the BES, based on the suitable economic optimal operation and environmental optimal operation, we choose 4.2 megawatt-hour BES to be used in the proposed BEMS. This is because this size of BES provides the minimum *TOC*, and the *TCOE* is slightly higher than the minimum value. The system parameters of BEMS used for the simulation are shown in Table 2.

**Table 2.** System parameters.

| Description | Notation | Value |
|---|---|---|
| CHP | | |
| Rated power (MW) | - | 24 |
| Electrical energy efficiency (%) | $\eta_{CHP,EE}$ | 33.9 |
| Power-to-heat ratio | P2H | 0.9244 |
| Maximum power production (MW) | $P_{CHP,max}$ | 24 |
| Minimum power production (MW) | $P_{CHP,min}$ | 4.8 |
| Electrical energy ramp rate (MW) | $R_{CHP}$ | 24 |
| $CO_2$ emission factor ($tCO_2$/MWh) | $EF_{CHP}$ | 0.5349 |
| Auxiliary Boiler | | |
| Rated heat power (MW) | - | 13.1882 |
| Efficiency (%) | $\eta_{AB}$ | 75 |
| Maximum heat production (MW) | $HP_{AB,max}$ | 13.1882 |
| Minimum heat production (MW) | $HP_{AB,min}$ | 2.6327 |
| $CO_2$ emission factor from natural gas combustion ($tCO_2$/MWh) | $EF_{AB}$ | 0.181 |
| Absorption Chiller | | |
| Rated cooling power | - | 42.2 |
| Coefficient of performance | $COP_{AC}$ | 1.1 |
| Maximum cooling production (MW) | $CP_{AC,max}$ | 42.2 |
| Minimum cooling production (MW) | $CP_{AC,min}$ | 8.44 |
| TES | | |
| Rated heat power (MW) | - | 50 |
| Heat charge rate (MW) | R1 | 15 |
| Heat discharge rate (MW) | R2 | 15 |
| Charging efficiency | $\varepsilon$ | 0.95 |
| Discharging efficiency | $\delta$ | 0.95 |
| Loss coefficient | $\mu$ | 0.001 |
| Initial heat energy in TES (MW) | init | 0–10 |
| Maximum heat storage (MW) | $S_{max}$ | 50 |
| Minimum heat storage (MW) | $S_{min}$ | 5 |
| BES | | |
| Charging efficiency | $\eta_c$ | 0.9 |
| Discharging efficiency | $\eta_d$ | 0.9 |
| Charge rate | $d_{sc}$ | 1.05 |
| Discharge rate | $d_{sd}$ | 1.05 |
| Inverter efficiency | $\eta_i$ | 1 |
| Max capacity (MWh) | - | 4.2 |
| Initial energy in BES (MW) | init_b | 0.84 |
| Maximum BES storage (MW) | $B_{max}$ | 3.78 |
| Minimum BES storage (MW) | $B_{min}$ | 0 |

We demonstrate the operating results of the shopping mall, which utilizes electricity from the 69-kV distribution grid. The electrical load profile has a range from 5 to 24 MW, and the cooling load profile has a range from 0 to 24 MW [16]. Therefore, a 24 MW *CHP* is applied with the proposed BEMS to match the load demand. The size of auxiliary boiler is chosen based on a suitability with *CHP* and the double-effect absorption chiller. Considering capacity of boiler, the main source of heat energy are *CHP* and TES. Therefore, the size of the boiler is chosen to accommodate the excess heat supplied by the main energy source [17]. The overall guidelines for choosing the system parameters are based on the appropriate capacity with demand matching.

*5.2. Economics Optimal Operation and Environmental Optimal Operation*

Economics optimal operation considers the *TOC* minimization. *TOC* represents the cost of operations in the energy dispatch. It consists of the energy consumption of the equipment in the BEMS. From the numerical experiment and comparison with the previous BEMS, it was found that the proposed BEMS can help reduce *TOC* by 9.68% for economics

optimal operation under the uncertainty of load demand and 1.26% for economics optimal operation under the nominal load case. Tables 3 and 4 summarize the *TOC* for the uncertainty of load demand and nominal load case, respectively. We observe the *TOC* classified for the main equipment of BEMS. It can be explained that the supplying load demand by BES reduces the purchase of grid power and cuts the demand charge cost.

**Table 3.** Economics optimal operation for uncertainty case.

| Objective Function | Previous BEMS | Proposed BEMS | Improvement |
|---|---|---|---|
| TOC (baht) | 6,662,941 | 6,018,029 | 9.68 |
| | TOC Classified by elements | | |
| CHP (baht) | 5,426,078 | 5,443,426 | −0.32 |
| Auxiliary boiler (baht) | 474,893 | 474,893 | 0 |
| BES (baht) | 0 | 48,731 | N/A |
| Power grid (baht) | 761,970 | 0 | 100 |
| SR (baht) | 0 | 99,711 | N/A |

**Table 4.** Economics optimal operation for nominal case.

| Objective Function | Previous BEMS | Proposed BEMS | Improvement |
|---|---|---|---|
| TOC (baht) | 6,050,087 | 5,973,810 | 1.26 |
| | TOC Classified by elements | | |
| CHP (baht) | 5,467,972 | 5,474,081 | −0.11 |
| Auxiliary boiler (baht) | 474,893 | 474,893 | 0 |
| BES (baht) | 0 | 48,731 | N/A |
| Power grid (baht) | 107,221 | 24,836 | 76.84 |

TCOE is the emission of carbon dioxide in the energy dispatch. This includes the carbon dioxide emission of the equipment in the BEMS. The results in Table 5 show that the proposed BEMS can reduce total carbon dioxide emission by 0.25% for environmental optimal operation under load uncertainty. To further analyze the amount of reduced emission, classified by the equipment in the BEMS, the dispatch of BES reduces the import of electricity from the power grid and clearly reduces the carbon dioxide emission. In contrast, Table 6 shows that applying the proposed BEMS to the nominal case, the total carbon emission is increased by 0.71%.

**Table 5.** Environmental optimal operation for uncertainty case.

| Objective Function | Previous BEMS | Proposed BEMS | Improvement |
|---|---|---|---|
| TCOE (tonCO$_2$) | 1569 | 1565 | 0.25 |
| | TCOE Classified by elements | | |
| CHP (tonCO$_2$) | 1360 | 1381 | −1.54 |
| Auxiliary boiler (tonCO$_2$) | 162 | 162 | 0 |
| BES (tonCO$_2$) | 0 | 12 | N/A |
| Power grid (tonCO$_2$) | 31 | 0 | 100 |
| SR (tonCO$_2$) | 0 | 23 | N/A |

**Table 6.** Environmental optimal operation for nominal case.

| Objective Function | Previous BEMS | Proposed BEMS | Improvement |
|---|---|---|---|
| TCOE (tonCO$_2$) | 1545 | 1556 | −0.71 |
| | TCOE Classified by elements | | |
| CHP (tonCO$_2$) | 1339 | 1383 | −3.30 |
| Auxiliary boiler (tonCO$_2$) | 163 | 172 | −4.91 |
| BES (tonCO$_2$) | 0 | 12 | N/A |
| Power Grid (tonCO$_2$) | 3 | 1 | 66.67 |

We present an example load profile with load uncertainty. Figure 13 shows the electrical energy flow to energy load under economics optimal operation, as well as a comparison between the previous and proposed BEMS. It was found that, when load uncertainty arises, the previous BEMS supplies the uncertain load by importing the electrical energy from power grid, while the proposed BEMS supplies the uncertain load by SR. In Figure 14, *CHP* plays a key role in suppling heat energy for absorption chiller, and TES shall supply heat energy when the demand is high. According to Figure 15, the state-of-charge of TES displays that the waste heat from *CHP* is stored to TES, and it is discharged to the absorption chiller when the cooling load demand is high. In Figure 16, the state-of-charge of BES illustrates that BES is fully charged by the first four hours of the day, and the energy is discharged to support the uncertainty during on-peak period. In Figure 17, the previous BEMS has imported energy from the power gird, whereas the proposed BEMS does not import energy from the power grid. The installation of BES can support the load uncertainty and does not require the imported energy from the power grid.

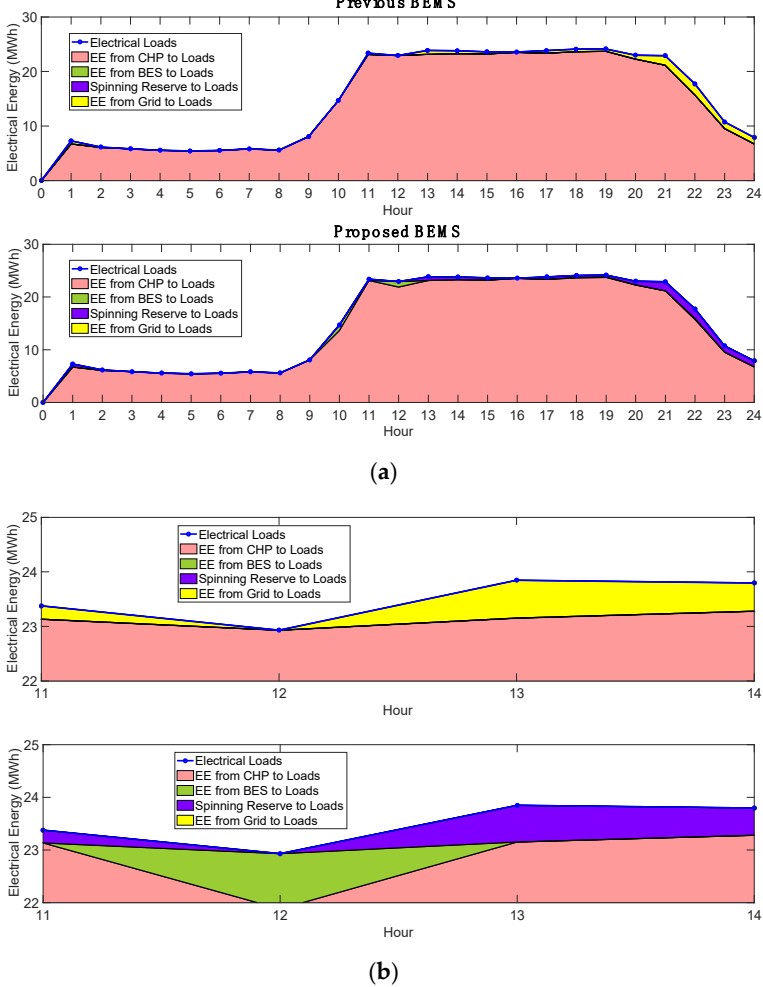

**Figure 13.** (**a**) Comparison of electrical energy flow to electrical load between the previous and proposed BEMS under economics optimal operation (uncertainty case). (**b**) Enlarged comparison of electrical energy flow to electrical load between the previous and proposed BEMS under economics optimal operation (uncertainty case).

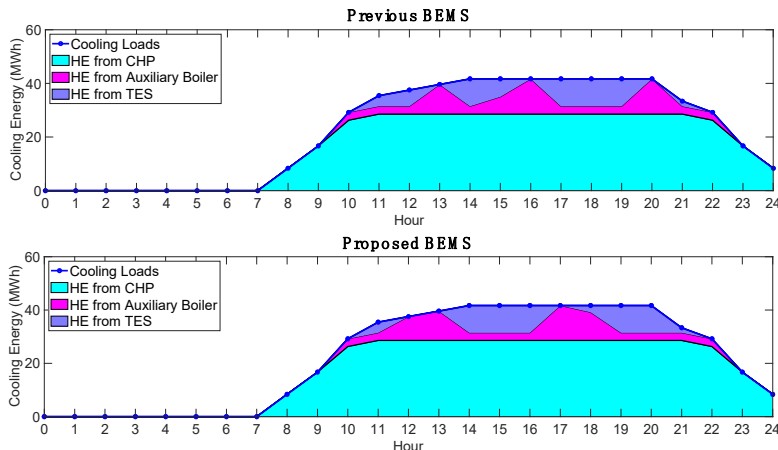

**Figure 14.** Comparison of cooling energy flow to cooling load between the previous and proposed BEMS under economics optimal operation (uncertainty case).

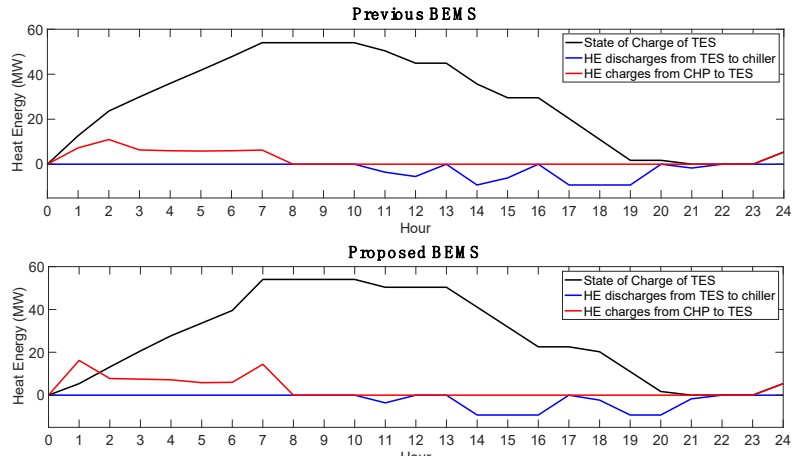

**Figure 15.** Comparison of state-of-charge of TES between the previous and proposed BEMS under economics optimal operation (uncertainty case).

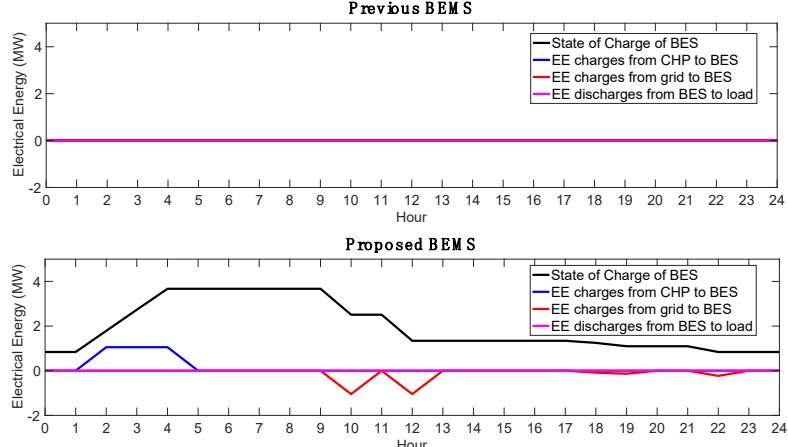

**Figure 16.** Comparison of state-of-charge of BES between the previous and proposed BEMS under economics optimal operation (uncertainty case).

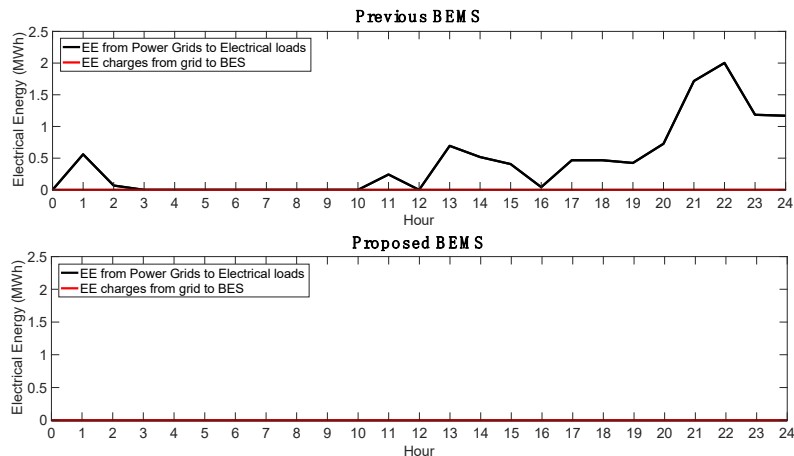

**Figure 17.** Comparison of power grid purchased between the previous and proposed BEMS under economics optimal operation (uncertainty case).

### 5.3. Multi-Objective Optimal Operation

From optimal energy dispatch strategies provided in Section 4.1, we utilize the multi-objective operation of BEMS with a 4.2 megawatt-hour BES and vary a weight factor from 0 to 1, with an increment of 0.1. When the weight is 0, only the optimal economic operation is considered. When the weight is 1, only the environmental optimal operation is considered. The experiments are divided into two cases. The first case is referred to as the uncertainty case, where we consider the predicted load with load uncertainty. The second case is referred to as the nominal case, where we consider the predicted load without load uncertainty. The results of the *TOC* and *TCOE* are obtained from the multi-objective optimal operation, where the cost function is linear combination of the normalized *TOC* and normalized *TCOE*. According to the minimum and maximum values in Equations (9) and (10), the maximum and minimum values of *TOC* and *TCOE* are shown in Table 7.

**Table 7.** Maximum and minimum value of *TOC* and *TCOE* for normalization.

| Value | Uncertainty Case | | Nominal Case | |
|---|---|---|---|---|
| | TOC (Baht) | TCOE (tonCO$_2$) | TOC (Baht) | TCOE (tonCO$_2$) |
| Minimum | 6,018,029 | 1565 | 5,973,810 | 1556 |
| Maximum | 6,333,777 | 1658 | 6,461,918 | 1659 |

From the results, when the weight factor is equal to 0, the normalized *TOC* is 0, which corresponds to the minimum *TOC*. *TOC* tends to increase as the weight increases. When the weight is equal to 1, *TOC* is the maximum value, and the normalized *TOC* is equal to 1 for both uncertainty and nominal cases. Moreover, when the weight is 0, the normalized *TCOE* is 1, which corresponds to the maximum *TCOE*. The *TCOE* tends to decrease as the weight increases. In particular, when a weight is 1, we obtain the minimum *TCOE*, and the normalized *TCOE* is equal to 0 for both uncertainty and nominal cases. The relationships shown in Figures 18 and 19 display a trade-off performance between *TOC* and *TCOE*.

The relationship between economics optimal operation and environmental optimal operation allows the user to determine the operating point of the system. For example, we can select a weight equal to 1 for the uncertainty case. When the proposed BEMS operates with low *TOC*, it results in high *TCOE*. Therefore, the curve is useful for the users to determine the desired operation point.

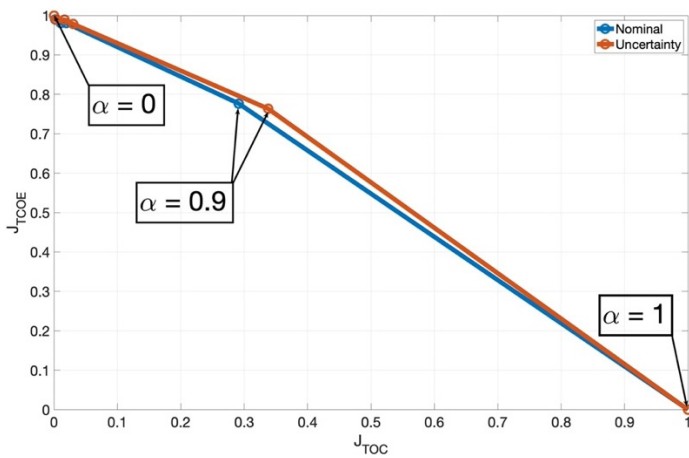

**Figure 18.** Trade-off performance between *TOC* and *TCOE*.

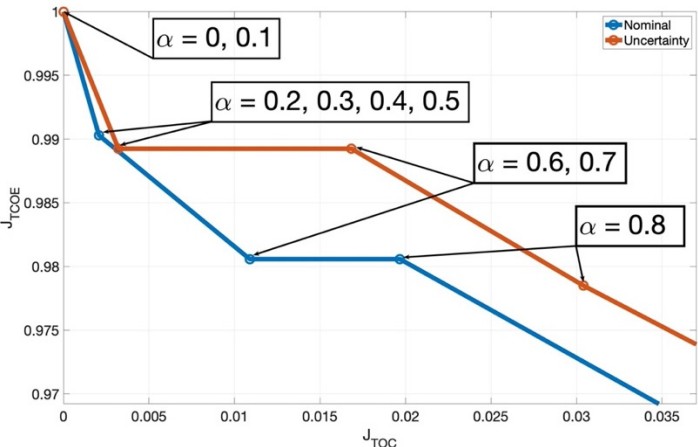

**Figure 19.** Enlarged trade-off performance between *TOC* and *TCOE*.

Next, we present the results of the proposed BEMS under multi-objective optimal operation. We compare the results of the proposed BEM (with BES and SR) and previous BEMS (without BES and SR) [18]. We depict the multi-objective optimal operation with weight equal to 0.9 and 1 and compare it to the previous BEMS. The results of *TOC* and *TCOE* are provided in Tables 8 and 9, respectively. The multi-objective optimal operation can reduce *TOC* by 7.33% and 4.94% for weights of 0.9 and 1, respectively. For weight equal to 1, it reduces *TCOE* by 0.25%; however, for weight 0.9, it increased *TCOE* by 4.27%. Note that BES and SR are clearly utilized to reduce the imported electricity from the power grid.

**Table 8.** TOC when the weight is 0.9 and 1.

| Objective Function | Previous BEMS | Multi-Objective Function | | Improvement | |
|---|---|---|---|---|---|
| | | $\alpha = 0.9$ | $\alpha = 1$ | $\alpha = 0.9$ | $\alpha = 1$ |
| TOC (baht) | 6,662,941 | 6,174,528 | 6,333,777 | 7.33 | 4.94 |
| | | TOC Classified by elements | | | |
| CHP (baht) | 5,426,078 | 5,418,901 | 5,481,719 | 0.13 | −1.03 |
| Auxiliary boiler (baht) | 474,893 | 571,756 | 755,313 | −20.4 | −59.05 |
| BES (baht) | 0 | 48,731 | 48,731 | N/A | N/A |
| Power grid (baht) | 761,970 | 0 | 0 | 100 | 100 |
| SR (baht) | 0 | 85,347 | 96,744 | N/A | N/A |

**Table 9.** TCOE when the weight is 0.9 and 1.

| Objective Function | Previous BEMS | Multi-Objective Function | | Improvement | |
|---|---|---|---|---|---|
| | | $\alpha = 0.9$ | $\alpha = 1$ | $\alpha = 0.9$ | $\alpha = 1$ |
| TCOE ($tonCO_2$) | 1569 | 1636 | 1565 | –4.27 | 0.25 |
| | | TOC Classified by elements | | | |
| CHP ($tonCO_2$) | 1360 | 1481 | 1381 | –8.9 | –1.58 |
| Auxiliary boiler ($tonCO_2$) | 162 | 122 | 162 | 24.7 | 0.16 |
| BES ($tonCO_2$) | 0 | 12 | 12 | N/A | N/A |
| Power grid ($tonCO_2$) | 31 | 0 | 0 | 100 | 100 |
| SR ($tonCO_2$) | 0 | 21 | 23 | N/A | N/A |

We present a power flow of the multi-objective function of BEMS with TES and BES. We show the power flow of the proposed BEMS, with a 4.2 megawatt-hour BES in presence of the load uncertainty and compare the weights of 0.9 and 1. Figure 20 illustrates the electrical energy flow to the electrical load. For both weights, we can completely support the load demand. Electrical energy is charged to BES, and we can make profits by selling electricity to the power grid. Figure 21 shows the flow of the thermal energy to cooling load. It can be seen that TES can reduce the need for auxiliary boiler. In the case that the weight is equal to 1, the thermal energy from the auxiliary boiler is used at certain periods and replaces the use of thermal energy from *CHP*. The reason is that the auxiliary boiler has a lower carbon dioxide emission rate. Figure 22 depicts the state-of-charge of TES. The heat waste is stored in TES and supply thermal energy to the absorption chiller at different periods. Figure 23 depicts the state-of-charge of BES. It can be seen that the electricity is charged to BES in the first four hours; then, it supplies the electrical load at different periods.

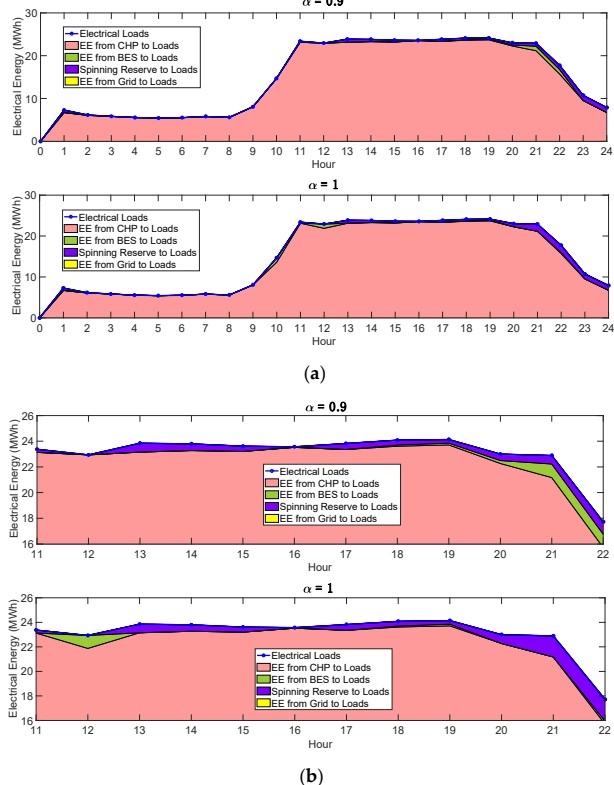

**Figure 20.** (**a**) Electrical energy flow to electrical load when the weight is 0.9 and 1. (**b**) Enlarged electrical energy flow to electrical load when the weight is 0.9 and 1.

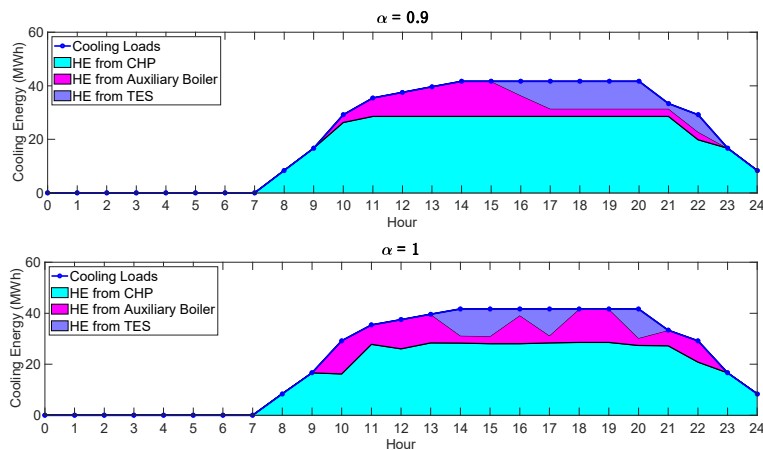

**Figure 21.** Cooling energy flow to cooling load when the weight is 0.9 and 1.

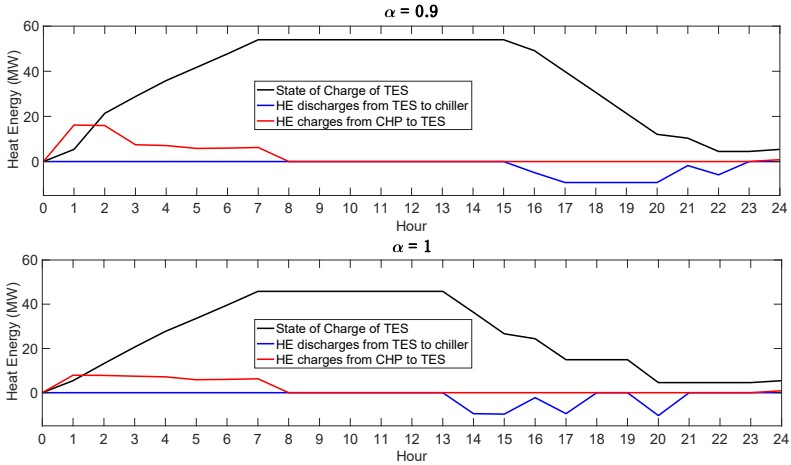

**Figure 22.** State-of-charge of TES when the weight is 0.9 and 1.

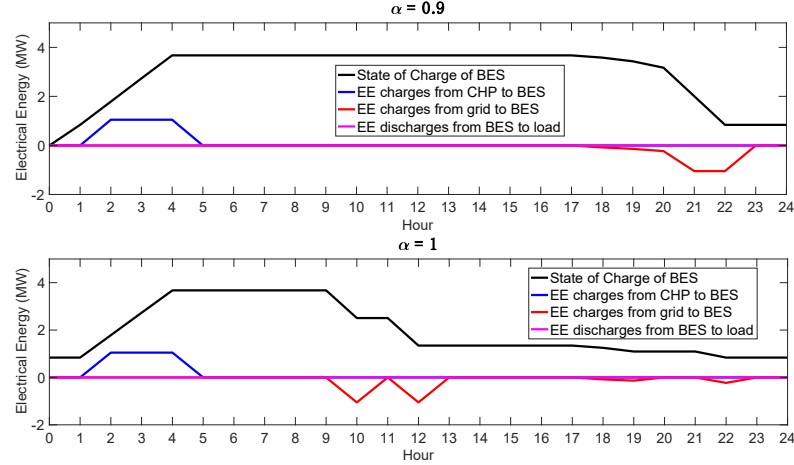

**Figure 23.** State-of-charge of BES of weighting factor 0.9 and 1.

## 6. Conclusions

This paper proposes the optimal dispatch strategies of BEMS with cogeneration, TES, and BES, considering load demand uncertainty. The objectives are two-fold. The first objective is the economics optimal operation, which aims to minimize the *TOC*. The second objective is the environmental optimal operation, which aims to minimize the *TCOE*. Afterward, the two objectives are combined into the multi-objective optimal operation.

We present a prediction of the electrical load demand using the ANN, LSTM, and CNN models, followed by analysis of the load demand uncertainty. As a result, we obtain the predicted load and load demand uncertainty, which are used for planning the energy dispatch. We apply the proposed dispatch strategies of electrical and cooling energy to a large shopping mall. For the load uncertainty case, the proposed BEMS reduces *TOC* by 9.68% under economic optimal operation and reduces *TCOE* by 0.25% under environmental optimal operation. For the nominal case, the proposed BEMS reduces *TOC* by 1.26% under economic optimal operation. However, the *TCOE* under environmental optimal operation is slightly increased by 0.71%. Moreover, the results show that BES provides a significant contribution to cutting the imported electricity from the power grid.

For the multi-objective optimal operation, we propose the normalized *TOC* and *TCOE* and consider the cost function as a linear combination of the normalized *TOC* and *TCOE*. The results show the relationship of *TOC* and *TCOE* as a trade-off performance. It can help users decide the operating point of BEMS. For energy flow analysis, the multi-objective optimal operation can completely meet the energy demand, as well as cut the import of electricity from the power grid. For the weight of 0.9, the proposed BEMS reduced *TOC* by 7.33%, whereas *TCOE* is increased by 4.2%.

Considering the rated power of *CHP*, it can be seen that the increase of efficiency of *CHP* clearly decreased *TOC* and *TCOE*. Likewise, the increase of efficiency of auxiliary boiler decreased *TOC* and *TCOE* from the boiler. Moreover, the other important factor is the *P2H* that represents the ratio between electricity from cogeneration and useful heat, when operating in full cogeneration mode. The cogeneration system works most efficiently when the *P2H* of the cogeneration system is close to the *P2H* of the building. Therefore, system parameters affect the results of *TOC* and *TCOE*.

**Author Contributions:** Conceptualization, D.B.; methodology, D.B. and P.T.; software, P.T.; validation, D.B. and P.T.; formal analysis, D.B. and P.T.; investigation, D.B.; resources, D.B.; data curation, P.T.; writing—original draft preparation, P.T.; writing—review and editing, D.B.; visualization, P.T.; supervision, D.B.; project administration, D.B.; funding acquisition, D.B. All authors have read and agreed to the published version of the manuscript.

**Funding:** This research was funded by the Rachadapisek Sompote Fund of the Intelligent Control Automation of Process Systems Research Unit.

**Institutional Review Board Statement:** Not applicable.

**Informed Consent Statement:** Not applicable.

**Data Availability Statement:** Not applicable.

**Conflicts of Interest:** The authors declare no conflict of interest.

## Appendix A

**Table A1.** Variables of BEMS.

| Variables | Description |
| --- | --- |
| $x_1$ | Electrical energy flow from CHP to electrical load |
| $x_2$ | Electrical energy flow from CHP to power grid |
| $x_3$ | Electrical energy flow from power grid to electrical load |
| $x_4$ | Heat energy flow from CHP to absorption chiller |
| $x_5$ | Waste heat from CHP to TES |
| $x_6$ | Heat energy flow from auxiliary boiler to absorption chiller |

**Table A1.** *Cont.*

| Variables | Description |
|:---:|:---|
| $x_7$ | Cooling energy from absorption chiller to cooling load |
| $x_8$ | Heat energy flow from TES to absorption chiller |
| $x_9$ | State-of-charge of TES |
| $x_{10}$ | Electrical energy flow from CHP to BES |
| $x_{11}$ | Electrical energy flow from BES to electrical load |
| $x_{12}$ | State-of-charge of BES |
| $x_{13}$ | Electrical energy flow from power grid to BES |
| $x_{14}$ | Electrical energy flow from BES to power grid |
| $x_{15}$ | Electrical energy flow from SR to electrical load |

**Table A2.** List of nomenclature.

| | |
|:---:|:---|
| $U_k$ | Load demand |
| $\hat{U}_k$ | Predicted load |
| $\Delta U_k$ | Prediction error |
| $\Delta U_{k,max}$ | Worst prediction error |
| $\hat{U}_{wc,k}$ | Predicted load under load demand uncertainty |
| $C_k$ | Cooling load |
| $p_k$ | Electrical energy charging price from power grid (baht/kWh) |
| $q_k$ | Electrical energy selling price to power grid (baht/kWh) |
| $C_{AB}$ | Operating cost of auxiliary boiler (baht/kWh) |
| $d_{PG}$ | Demand charge of imported power (baht/MW) |
| $\Delta t$ | Time duration of time interval |
| $C_{CHP}$ | Operating cost of CHP (baht/kWh) |
| $n$ | The number of operating hours |
| $d$ | The number of operating days |
| $EF_{CHP}$ | $CO_2$ emission factor of CHP (tonCO$_2$/MWh) |
| | $CO_2$ emission factor of boiler (tonCO$_2$/MWh) |
| $GEF$ | Power grid emission factor (tonCO$_2$/MWh) |
| $\eta_{AB}$ | Efficiency of auxiliary boiler |
| $\varepsilon$ | Charging efficiency of TES |
| $\delta$ | Discharging efficiency of TES |
| $R_1$ | Heat charging rate |
| | Heat discharging rate |
| $\mu$ | Loss coefficient of TES |
| $S_{min}$ | Minimum capacity of TES |
| $S_{max}$ | Maximum capacity of TES |
| $d_{sc}$ | Charging rate of BES |
| $d_{sd}$ | Discharging rate of BES |
| $\eta_c$ | Charging efficiency of BES |
| $\eta_i$ | Charging efficiency of inverter |
| $B_{min}$ | Minimum capacity of BES |
| $B_{max}$ | Maximum capacity of BES |
| $P_{CHP,min}$ | Minimum capacity of CHP |
| $P_{CHP,max}$ | Maximum capacity of CHP |
| $P2H$ | Power-to-heat ratio |
| $R_{CHP}$ | Ramp rate of CHP |
| $CP_{AC,min}$ | Minimum capacity of AC |
| $CP_{AC,max}$ | Maximum capacity of AC |
| $COP_{AC}$ | Efficiency coefficient of AC |
| $H_{AB,min}$ | Minimum capacity of AB |
| $H_{AB,max}$ | Maximum capacity of AB |
| $J_{TOC}$ | Normalized TOC |
| $J_{TCOE}$ | Normalized TCOE |
| $\alpha$ | Weighting factor |

**Table A3.** List of acronyms.

| | |
|---|---|
| AB | Auxiliary boiler |
| AC | Absorption chiller |
| ANN | Artificial neural network |
| BEMS | Building energy management system |
| BES | Battery energy storage |
| CHP | Combined heat and power |
| CNN | Convolutional neural network |
| LSTM | Long short-term memory |
| MAPE | Mean absolute percentage error |
| RMSE | Root mean squared error |
| SOC | State-of-charge |
| SR | Spinning reserve |
| TCOE | Total carbon dioxide emission |
| TES | Thermal energy storage |
| TOC | Total operating cost |

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
