# Peer review of "Multi-Objective Optimal Operation of Building Energy Management Systems with Thermal and Battery Energy Storage in the Presence of Load Uncertainty"

_sustainability, doi:10.3390/su141912717_

Round 1
Reviewer 1 Report
The manuscript was well organized and the content about the optimal operation of BEMS with TES and BES considering load demand uncertainty aims to reduce total operating cost (TOC) and total CO2 emission (TCOE) is beneficial and draw attention to the field.
Manuscript presents valuable detailed method for the validation of ANN-LSTM-CNN load demand forecasting and uncertainty and then apply the optimal dispatch strategies considering both economics and environmental optimal operation considered as the important objectives.
The manuscript was written constructively which is easy for readers to follow, although some minor typos are found in the manuscript. The manuscript is recommended to be published after some minor corrections as follow.
1. Some minor typos and some minor grammatical errors are found in the manuscript. A grammar check may be required once again., e.g. in line51 of section1 Likewise, [6] proposed the energy dispatch strategy to schedule the charging and discharging of the battery of the battery-integrated power system., in line74 the conclusion are given in section 6., in line 405 of section 4.3. dexcribed
2. Figure 9 and Figure 10, numbers shown on X-Y axis are difficult to see.
3. Figure 13 and Figure 20 especially numbers on the upper-left corner are difficult to see. The bigger figure with bigger font size may help.
4. Is there any specific reason that the authors selected the system parameter to find the result in Table 2 such as the CHP rated power at 24 MW or TES rated power at 50 MW while Auxiliary Boiler rated heat power is quite lower at 13.1882 MW? If the authors mention additional information on the guideline of choosing the system parameter in the study, this will make the readers better understand and the manuscript will be more attractive.
5. Some more discussion (in conclusions section) on how TOC/TCOE results with system behavior correlatedly respond on changing system parameter i.e., rated power of CHP, Auxiliary Boiler with higher or lower efficiency.
Reviewer 2 Report
1. Abstract: the first sentence has grammer mistake. There are a few significant grammer mistakes found in Abstract, which has severely affected the reading experience. Please pay attention to the grammer of the manuscript and have professional English editor to edit the English of the manuscript.
2. Please further elaborate the development of proposed BEMS.
3. The authors used machine learning methods to predict the optimal size of BES by considering economics and 14 environmental optimal operation. The authors should address the source of the numerical data for processing the machine learning.
4. The authors have elaborated the modelling and model selections in details while the discussion on the results are absent. What does the prediction lead to the implication of practice? The authors should address this part clearly.
Reviewer 3 Report
The paper “Multi-Objective Optimal Operation of Building Energy Management Systems with Thermal and BES in the Presence of Load Uncertainty” (authors: P. Trairat and D. Banjerdpongchai) has been reviewed.
There are some suggestions and comments:
1. The abbreviations' chapter must be added to the text. It is better not to describe the abbreviations in the abstract of the paper.
Were the last propositions of the abstract generated in the research?
2. It is better to avoid the words “many” in the description of energy types such the electricity and thermal energy.
3. The literature sources should be cited in these sentences: “Many of research reports [??] load demand prediction and takes the load demand forecast error into consideration in planning the power distribution. The prediction error is classified in the form of uncertainty [??].”
4. Where is the heating load in figure 1? Is the research based on a specific object and loads? It should be explained.
5. One universal figure with n neurons may be presented instead of figures 2 and 3.
6. What are the MAPE and RMSE? It needs to be explained in the text or in the abbreviation chapter.
7. It would be better to divide figure 13 into two figures.
8. Reading the text, it remains unclear for which buildings and under what conditions the proposed improvements can be applied. How universal is it? And so on.
Conclusion
The results are interesting, and I think this paper can be accepted for publication in the journal after major revision.
Author Response
There are only two reviewers.

Round 2
Reviewer 3 Report
The revised version of the article “Multi-Objective Optimal Operation of Building Energy Management Systems with Thermal and BES in the Presence of Load Uncertainty” (authors: P. Trairat and D. Banjerdpongchai) and the authors’ responses have been reviewed.
It can be noticed that the authors revised, corrected and added additional information to the paper. The manuscript was improved.
Author Response
Thank you for your thoughtful comments. We edited the manuscript and corrected the grammatical errors and adjusted the layout. Now the second revision of manuscript is ready for your consideration. If there is any additional comment or suggestion, please let us know.